# Dietary Supplementation of Ancientino Ameliorates Dextran Sodium Sulfate-Induced Colitis by Improving Intestinal Barrier Function and Reducing Inflammation and Oxidative Stress

**DOI:** 10.3390/nu15122798

**Published:** 2023-06-19

**Authors:** Meng Liu, Yuhui Wang, Guoqiang Guan, Xi Lu, Yizhun Zhu, Xiaoqun Duan

**Affiliations:** 1School of Pharmacy, Faculty of Medicine, Macau University of Science and Technology, Macau SAR 999078, China; 21098533pp30001@student.must.edu.mo; 2School of Biomedical Industry, Guilin Medical University, Guilin 541199, China; 112016010@glmc.edu.cn (Y.W.); 21207041261@stu.glmc.edu.cn (G.G.); 3Industrial Technology Research Institute, Guilin Medical University, Guilin 541199, China; 4School of Pharmacy, Guilin Medical University, Guilin 541199, China; 122006033@glmc.edu.cn; 5State Key Laboratory of Quality Research in Chinese Medicine, School of Pharmacy, Macau University of Science and Technology, Macau SAR 999078, China

**Keywords:** ulcerative colitis, Ancientino, inflammatory response, intestinal barrier function, oxidation stress

## Abstract

Ancientino, a complex dietary fiber supplement mimicking the ancient diet, has improved chronic heart failure, kidney function, and constipation. However, its effect on ulcerative colitis is unknown. This study explores the impact of Ancientino on colitis caused by dextran sulfate sodium (DSS) and its mechanisms. Data analyses showed that Ancientino alleviated bodyweight loss, colon shortening and injury, and disease activity index (DAI) score, regulated levels of inflammatory factors (tumor necrosis factor-alpha (TNF-α), interleukin-10 (IL-10), interleukin-1 beta (IL-1β), and interleukin 6 (IL-6)), reduced intestinal permeability (d-lactate and endotoxin), fluorescein isothiocyanate–dextran (FITC-dextran), and diamine oxidase (DAO), repaired colonic function (ZO-1 and occludin), and suppressed oxidative stress (superoxide dismutase (SOD), catalase (CAT), glutathione peroxidase (GSH-Px), and malondialdehyde (MDA)) in vivo and in vitro. In short, this study demonstrated that Ancientino alleviates colitis and exerts an anticolitis effect by reducing inflammatory response, suppressing oxidative stress, and repairing intestinal barrier function. Thus, Ancientino may be an effective therapeutic dietary resource for ulcerative colitis.

## 1. Introduction

Inflammatory bowel diseases (IBD), including ulcerative colitis (UC) and Crohn’s disease (CD), are chronic inflammatory intestinal disorders that are characterized by weight loss, abdominal pain, diarrhea, and fecal occult blood [1]. It is estimated that millions of people are suffering from IBD worldwide, and the number is increasing [2]. Although the precise etiology of IBD is not fully understood, a consensus exists that environmental, genetic, immunological, and infectious factors are triggering its complex pathogenesis [3]. Moreover, diet is now considered the most important environmental factor affecting the incidence of UC [4]. Accumulating evidence suggests that the Western dietary pattern, defined as a high sugar and fat/low-fiber diet, carries a greater risk of developing UC than the Mediterranean or high-fiber ketogenic diet eating pattern [5]. Therefore, improving dietary habits is considered a crucial strategy for managing UC.

IBD is associated with the impairment of colonic barrier function, the secretion of inflammatory cytokines, and excessive oxidative stress [6,7]. Accumulating clinical and experimental data has established these changes in UC patients and colitis mouse models [8]. The use of clinically available agents (e.g., 5-aminosalicylic acid and glucocorticoids) has not led to effective treatments because of various adverse effects and limitations [9]. Thus, food-assisted therapy for ulcerative colitis is receiving increased interest worldwide [10]. Polyphenols in a diet may significantly reduce colitis symptoms by inhibiting colon inflammation and improving intestinal barrier health [11]. As a consequence of dietary fiber supplementation, the intestinal barrier can also remain intact during intestinal digestion [12]. Moreover, existing research suggests that edible berries (e.g., strawberry, maqui berry, blueberries, and black currant) could interfere with inflammatory transduction pathways and diminish oxidative stress, thus protecting mice from dextran sulfate sodium (DSS)-caused gut inflammation [13,14]. Ancientino, a complex dietary fiber supplement that mimics the ancient diet, has improved chronic heart failure, kidney function, and chronic constipation. However, its effect on ulcerative colitis is unknown.

Based on the above findings, this study described the anti-UC effect of Ancientino and the underlying mechanisms in terms of the inflammatory response, colonic barrier function, and oxidation stress. It offered a solid basis for the use of Ancientino as a functional food in the adjuvant therapy of UC.

## 2. Materials and Methods

### 2.1. Materials

DSS (S5036, 36–50 kDa) was from MPBio (Solon, OH, USA). Fluorescein isothiocyanate (FITC)-dextran (60842-46-8) of MW 3000–5000 kDa and lipopolysaccharide (LPS, L3129) were from Sigma-Aldrich (St. Louis, MO, USA). Tumor necrosis factor-alpha (TNF-α, CT303A), interleukin-10 (IL-10, CT307A), interleukin 6 (IL-6, CT299A), and interleukin-1 beta (IL-1β, 1210122) enzyme-linked immunosorbent assay (ELISA) kits were supplied by Dakowei Biotech Co., Ltd. (Beijing, China). Myeloperoxidase (MPO, A044-1-1), malondialdehyde (MDA, A003-1), catalase (CAT, A007-2), superoxide dismutase (SOD A001-3), and glutathione peroxidase (GSH-Px, A005) assay kits were from Nanjing Jiancheng Bioengineering Company (Nanjing, China). The concentrations of serum diamine oxidase (DAO, CSB-E10090m) and endotoxin (CSB-E13066m) ELISA kits were from Cusabio LLC. (Houston, TX, USA). The d-lactate (JL48176) ELISA Kit was obtained from Shanghai Future Industrial Limited by Share Ltd. (Shanghai, China). FastKing gDNA Dispelling RT Super Mix (KR118) and SYBR Green (FP205) were from TIANGEN Co., Ltd. (Beijing, China). Main primary antibodies: rabbit monoclonal antibodies recognizing ZO-1 (AF5145) and occludin (DF7504). They were provided by Affinity Biosciences (Cincinnati, OH, USA). Dilutions used were 1:1000 for Western blots and 1:200 for immunofluorescence experiments. Mouse monoclonal antibodies recognizing GAPDH (1:1000, TA-08) were from Zhongshan Golden Bridge Biotechnology Co., Ltd. (Beijing, China) Goat antirabbit (A21020) and antimouse (A21010) immunoglobulin G secondary antibodies (1:10,000) were from Abbikine, Inc. (Wuhan, China). Chemical agents were at least analytical grade.

### 2.2. Composition of the Ancientino

Ancientino was provided by Guilin Ancientino Food Technology Co., Ltd. (Guilin, China). The composite dietary fiber is composed of the following components in proportion to the weight percentage *w*/*w*:The ratio of xylan and glucomannan was 80%.The fructan and resistant starch ratio was 15%.Legume cotyledon fiber vitamin and potato fiber proportions were 5%.

Compound dietary fiber-specific components included plant cell wall hemicellulose’s primary component, wood poly sugar, glucomannan, and the second largest component of the plant cell wall, including hemicellulose and cereal nonstarch storage polysaccharide, in addition to the appropriate amount of resistant starch, legume cotyledon fiber, and potato fiber. These fibers are the representative fiber components of human staple foods and provide a large amount of fiber intake in the daily diet.

### 2.3. Animal Model Establishment and Treatment

C57BL/6 male mice (~20 g, 6–8 weeks) were from Selleck Jingda Experimental Animal Co., Ltd. (Changsha, China). These mice were kept individually under stable conditions with a 12 h:12 h light:dark cycle, controlled temperature, humidity, and always free access to their regular food and water. The bedding was replaced three times a week. Colitis was induced by adding 2.5% DSS (diluted with water) for seven days, followed by three days of water.

To examine the potential impact of Ancientino on colitis, 35 mice were randomly assigned to 5 groups (*n* = 7): normal group, model group, and Ancientino groups (300, 600, and 1200 mg/kg, respectively). The corresponding Ancientino doses were orally administered to the mice for 10 consecutive days (once a day), while the mice of normal and model groups (DSS-treated) received deionized water by gavage, and 0.1 mL/10 g of administration volume was used for each mouse. On the 10th day, these mice were sacrificed with chloral hydrate, and eyeballs were taken for blood. Then, their colons were removed and measured.

### 2.4. Disease Activity Index (DAI) Score

The mice’s weight, stool consistency, and stool bleeding were tracked and documented daily during the trial. The DAI was expressed as an average score of the three parameters [15]. Seen Table 1.

### 2.5. MPO Activity

Colon tissues were homogenized in ice-cold 0.9% NaCl (1:9, *w*/*v*) to obtain a 10% homogenate suspension. The MPO activity assay was conducted following the manufacturer’s protocols, and its activity was expressed in units per gram of tissue.

### 2.6. Histological Analysis

Fixation of colon tissue (48 h) with 4% paraformaldehyde was performed and embedded in paraffin. The embedded wax block was fixed on the microtome, cut into 5 μm thin slices, and later stained using hematoxylin and eosin (H&E), capture on camera under a microscope. As previously mentioned, double-blinded pathologists performed histological grading to gauge the degree of colonic inflammation. Table 2 displays the histological scoring criteria. The histology score is the sum of the blow indices (maximum score: 10).

### 2.7. Intestinal Permeability Analysis In Vivo

The serum FITC-dextran was utilized to assess intestinal permeability in vivo. Briefly, after being fed with DSS for seven days, the mice were fasted for 4 h and then orally administered 0.3 mL FITC-Dextran (6 mg/10 g). After three hours, blood was collected from each mouse’s orbital sinuses. The blood was centrifuged at 3500 rpm for 10 min at 4 °C. Then, 0.1 mL of serum was detected using a Synergy HT plate reader (BioTek, Winooski, VT, USA). The fluorescence intensity of the serum (excitation at 485 nm; emission at 535 nm) was used to determine the level of intestinal permeability. Serum FITC-dextran concentrations were calculated from a standard curve of serially diluted FITC-dextran in PBS.

### 2.8. Determination of DAO, Endotoxin, and d-Lactate

Serum samples were obtained by collecting blood and subjecting it to centrifugation at 3500 rpm for 10 min at 4 °C. The levels of DAO, endotoxin, and d-lactate were then measured using ELISA kits as directed by the manufacturer.

### 2.9. Caco-2 Cell Culture and Administration

The Caco-2 cell line was from the American Type Culture Collection (Manassas, VA, USA). It has been passed down and cultivated by members of our research group. Caco-2 cells were grown in Dulbecco’s modified Eagle’s medium (DMEM, Gibco, Gaithersburg, MD, USA) with 10% fetal bovine serum (FBS), 100 IU/mL penicillin, and 100 μg/mL streptomycin. The cells were maintained in a 5% CO_2_ incubator at 37 °C until they adhered to the culture dish. After a 6 h starvation period, the cells were cotreated with different doses of Ancientino (25, 50, 100 μg/mL) and LPS (1 μg/mL) for 24 h. Details about the assessment of cell viability are available in the Appendix A.

### 2.10. Immunofluorescence (IF)

After treating the Caco-2 cells with LPS (1 μg/mL), Ancientino (25, 50, and 100 μg/mL) was added and incubated for 24 h. The cells were treated with a primary antibody at 4 °C overnight after being incubated with the suspension solution (TBST was added with 5% nonfat milk) for 30 min at room temperature. A secondary antibody was added after four PBS washes, and the cells were then cultivated for an hour at room temperature. The counterstaining of DAPI was applied for 10 min following another wash. Using an Olympus microscope (Tokyo, Japan), cells were visualized immediately following the addition of an antifluorescence quenching agent.

### 2.11. Measurement of the Levels of Oxidative Stress Biomarkers

Colon tissues were washed and homogenized in ice-cold PBS before being centrifuged at 3500 rpm for 10 min at 4 °C to produce the colonic supernatant. The supernatant from Caco-2 cells was collected after 24 h of LPS stimulation, and MDA, CAT, GSH-Px, and SOD levels were individually measured using a detection kit.

### 2.12. Cytokine Measurement

Colon tissues or Caco-2 cells were extracted using lysis buffer and centrifuged at 12,000 rpm for 15 min at 4 °C. The protein supernatant was taken and quantified with the BCA Protein Assay Kit. The expression levels of inflammatory cytokines were individually detected following the manufacturer’s protocol.

### 2.13. Real-Time Reverse Transcription–Polymerase Chain Reaction (qRT-PCR)

Total RNA was extracted from colonic tissues or Caco-2 cells using Trizol reagent (Invitrogen, Carlsbad, CA, USA). Briefly, weigh about 40 mg of colonic tissue into an enzyme-free tube and add Trizol (1 mL) and two steel balls. Then, crush with a high-throughput tissue disruptor (60 Hz, 60 s, twice) until there are no visible colonic tissue fragments. For Caco-2 cells, discard the supernatant in the six-well plate and wash with precooled PBS twice. Add Trizol (1 mL) to each well and transfer to an enzyme-free tube. After this, the absorbance of each sample was measured by a NanoDrop 2000 Super Spectrophotometer (Thermo, Wilmington, DE, USA), and the corresponding RNA concentration was calculated. Subsequently, the above RNA was reverse transcribed using FastKing gDNA Dispelling RT Super Mix. Quantitative PCR was performed using SYBR Green in MyiQ2 Detection System (Bio-Rad, Hercules, CA, USA) with the following conditions: 95 °C, 10 s; 60 °C, 30 s. The expression levels of target genes were standardized to GAPDH as an internal reference, and data were analyzed using the 2^−ΔΔCt^ method. The primer sequence is seen in Table 3.

### 2.14. Western Blot (WB) Analysis

To prepare protein samples for WB analysis, portions of colon tissue or Caco-2 cells were lysed in RIPA buffers containing 1% PMSF and then grounded in the automatic sample grinding machine to collect the cracking solution. After being centrifuged at 12,000 rpm for 15 min at 4 °C, the supernatant was collected and quantified with a BCA kit. In order to acquire proteins for further investigation, the protein supernatant was thoroughly mixed with a loading buffer. Additionally, it was boiled in boiling water for 10 min to fully denature it. Then, the denatured protein samples were divided into several tubes and stored in a −20 °C refrigerator for use. In this experiment, according to the molecular weights of ZO-1 (195 kDa) and occludin (60 kDa), 10% separation glue and 5% concentrate glue were prepared for sodium dodecyl sulfate–polyacrylamide gel electrophoresis (SDS-PAGE) to separate the same amount (25 μg) of target proteins. After electrophoresis, the gel was transferred to the nitrocellulose filter (NC) membrane and sealed with fresh 5% skim milk for 1.5 h. Subsequently, the NC membranes were placed on a shaker and shaken at a constant speed twice for 6 min each time. Dilute the primary antibody according to the dilution ratio of the instructions, drain the water on the NC film, place them, respectively, in the antibody bag containing the primary antibody, drive away the bubbles, and incubate at 4 °C overnight. The next morning, take the antibody bag from the refrigerator and shake it in a shaker for 20 min. Then, take out the NC film and wash it with TBST 4 times for 7 min each time. Finally, the NC membrane was incubated with the corresponding secondary antibody (1:10,000) for 2 h. Enhanced chemiluminescence (ECL) WB detection reagents were utilized to visualize representative bands, and ImageJ software was used to quantify the band optical intensity.

### 2.15. Statistical Analysis

GraphPad Prism 9.0 software was used to draw a quantification diagram and analyze the data, and the data were expressed as the mean ± standard error of the mean (SEM). Statistical differences were assessed by one-way analysis of variance (ANOVA), followed by Dunnett’s test when comparing multiple independent groups. *p*-values < 0.05 indicated statistical significance.

## 3. Results

### 3.1. Ancientino Ameliorated the Symptoms of DSS-Induced Colitis in Mice

The protective effect of Ancientino against colitis was evaluated by body weight, DAI score, colon length, and pathological symptoms. As shown in Figure 1, compared with the normal group, with continuous intake DSS, C57BL/6 mice gradually developed colitis with bloody stool, diarrhea, body weight loss, and shortened colon length, showing a significant increase in DAI score and MPO activity (*p* < 0.01). On day 10, oral administration of Ancientino (300, 600, and 1200 mg/kg) protected (*p* < 0.05) against body weight loss (1.15-, 1.23-, and 1.32-fold, Figure 1A), decreased the DAI score (20.6%, 44.4% and 50.8%, Figure 1B), prevented colon shortening (1.29-, 1.40-, and 1.50-fold, Figure 1C), decreased MPO activity (21.3%, 35.6%, and 44.0%, Figure 1D), and improved the pathological inflammation and crypt damage (16.7%, 40.5%, and 66.7%, Figure 1E,F) compared with that in the DSS-induced enteritis mice group, respectively. Especially, the high-dose Ancientino (1200 mg/kg, *p* < 0.01) group had the best effect. These results indicate that Ancientino might help prevent the occurrence and development of UC.

### 3.2. Ancientino Alleviated Colitis by Suppressing Inflammatory Response In Vivo

The pathogenesis of IBD is known to be influenced by inflammatory cytokines, which can significantly impact intestinal barrier function and histological and clinical symptoms [16]. To investigate the influence of Ancientino on the regulation of inflammatory cytokines, we used ELISA and qRT-PCR to detect the expression levels of TNF-α, IL-1β, IL-6, and IL-10 protein and mRNA in colonic tissues, respectively. As seen in Figure 2, relative to the normal group, DSS significantly (*p* < 0.01) decreased the protein and mRNA level of IL-10 cytokines (56.8% and 60.7%, respectively), while significantly (*p* < 0.01) increasing the expression of TNF-α (3.47- and 7.39-fold, respectively), IL-6 (3.06- and 9.52-fold, respectively), and IL-1β (1.66- and 6.83-fold, respectively) at their protein and mRNA levels, indicating that inflammation was induced in mice. Remarkably, Ancientino, especially at the high dose, effectively reversed the abnormal production of inflammatory cytokines (*p* < 0.01) and mitigated the acute responses, thereby showing its potential role in controlling the inflammation.

### 3.3. Ancientino Improved Colitis by Reducing Intestinal Permeability In Vivo

Damage to the integrity of the intestinal barrier leads to an increased serum level of FITC-Dextran, DAO, d-lactate, and endotoxin [17]. To investigate the effects of dietary Ancientino on intestinal permeability in DSS-induced enteritis mice, we detected these chemical indicators (Figure 3). In comparison with the normal group, the levels of four indicators all showed marked elevation in the model group, increasing by 12.50-, 1.49-, 3.05-, and 1.62-fold, respectively. These phenomena indicated that the intestinal barrier of the enteritis mice was strongly impaired and its permeability increased. In contrast, Ancientino (1200 mg/kg) significantly (*p* < 0.01) attenuated the serum FITC-Dextran (Figure 3A), DAO (Figure 3B), d-lactate (Figure 3C), and endotoxin (Figure 3D) levels by 56.3%, 32.1%, 32.4%, and 51.4%, respectively, in comparison with the model group. Interestingly, the low-dose Ancientino (18.77 ± 1.24, 300 mg/kg) did not regulate (*p* > 0.05) the effect of DAO compared with the model group (20.66 ± 1.08). The findings suggested that the effectiveness of Ancientino in treating UC was closely linked to its ability to strengthen intestinal barrier integrity.

### 3.4. Ancientino Improved Colitis by Preserving the Expression of Tight Junction (TJ) Proteins In Vivo

To prove that TJs play a significant role in colonic function, we investigated the effect of Ancientino on the expression of TJs in colitis mice. WB and qRT-PCR examined the protein and mRNA expression levels of ZO-1 and occludin. As seen in Figure 4, the levels of ZO-1 and occludin were significantly lower (*p* < 0.01) than the normal group. However, treatment with Ancientino (1200 mg/kg) led to a notable (*p* < 0.01) upregulation in ZO-1 (7.93- and 1.89-fold, respectively) and occludin (8.19- and 2.51-fold, respectively) at their protein and mRNA levels, in comparison with the model group. It thus seems that intestinal barrier protection may contribute to the anticolitis potential of Ancientino.

### 3.5. Ancientino Improved Colitis by Removing the Excessive Oxidative Stress In Vivo

To assess the potential antioxidant properties of Ancientino in DSS-induced enteritis mice, we evaluated the activities of SOD, GSH-Px, CAT, and MDA levels in colon tissues (Figure 5A–D). The colitis model group in mice showed a significant (*p* < 0.01) increase in MDA (1.82-fold) levels and a decrease in the activities of SOD (54.7%), CAT (35.9%), and GSH-Px (32.6%) compared with their respective normal groups. However, Ancientino’s (1200 mg/kg) intervention reversed these changes, restoring the MDA (25.4%) level and inhibiting SOD (1.63-fold), CAT (1.49-fold), and GSH-Px (1.37-fold) activity reduction compared with the model group. An interesting finding was that Ancientino (300 mg/kg) did not show an effect of treatment (*p* > 0.05), while Ancientino (19.59 ± 0.77, 1200 mg/kg) almost restored the expression of CAT to normal levels (20.48 ± 0.79, *p* < 0.01). These results suggest that the anti-UC effect of Ancientino relied on removing excessive oxidation stress.

### 3.6. Ancientino Alleviated Colitis by Suppressing the Inflammatory Response In Vitro

We also investigated the influence of Ancientino on the regulation of TNF-α, IL-1β, IL-6, and IL-10 in LPS-stimulated Caco-2 cells using ELISA and qRT-PCR. As seen in Figure 6, relative to the control groups, LPS significantly (*p* < 0.01) decreased the protein (56.1%) and mRNA (67.7%) levels of IL-10 cytokines, while significantly increasing the expression of TNF-α (2.31- and 4.44-fold), IL-6 (2.43- and 5.27-fold), and IL-1β (2.62- and 8.05-fold) at their protein and mRNA levels, respectively, indicating that inflammation was induced in Caco-2 cells. Remarkably, the Ancientino 100 μg/mL-treated group experienced a better therapeutic effect (*p* < 0.01), effectively reversing the abnormal secretion of inflammatory cytokines. These findings supported the preceding in vivo findings and further confirmed Ancientino’s anti-inflammatory effects.

### 3.7. Ancientino Improved Colitis by Preserving the Expression of TJ Proteins In Vitro

To prove that TJs play a significant role in colonic function in vitro experiments, we explored the effect of Ancientino on the expression of TJs in LPS-stimulated Caco-2 cells. WB and qRT-PCR examined the levels of ZO-1 and occludin. As seen in Figure 7, the protein and mRNA expression levels of ZO-1 and occludin were significantly (*p* < 0.01) reduced compared with the normal group. However, treatment with Ancientino (100 μg/mL) led to a notable (*p* < 0.01) upregulation in ZO-1 (3.74- and 2.45-fold) and occludin (4.46- and 1.61-fold) at their protein and mRNA levels, respectively, in comparison with the LPS-treated group.

To further corroborate the aforementioned findings, we also performed the ability of Ancientino to counteract the effects of LPS on the content of both TJ proteins in the Caco-2 monolayer with IF (Figure 7E,F). Not unexpectedly, LPS decreased the abundance of ZO-1 and occludin, thus compromising the barrier integrity. Treatment with Ancientino significantly increased the abundance of both TJ proteins localized on the cell surfaces. This effect became especially more marked with high doses of Ancientino. In addition, unlike the membrane localization observed with occludin, ZO-1 showed a more defuse distribution pattern, whereas its overall fluorescence intensity increased with Ancientino. These findings confirmed Ancientino’s ability to maintain intestinal barrier integrity.

### 3.8. Ancientino Improved Colitis by Removing the Excessive Oxidative Stress In Vitro

To study the antioxidant properties of Ancientino on LPS-stimulated Caco-2 cells, we evaluated the activities of oxidative stress biomarkers in Caco-2 cells (Figure 8A–D). Compared with the control group, the LPS-induced Caco-2 cells showed a significant (*p* < 0.01) increase in MDA levels by 2.70-fold and a decrease in the activities of SOD (61.8%), CAT (50.1%), and GSH-Px (48.2%). However, Ancientino’s (100 μg/mL) intervention reversed these changes, restoring the MDA levels by 28.6% and inhibiting SOD (2.00-fold), CAT (1.83-fold), and GSH-Px (1.48-fold) activity reduction relative to the model group, respectively. An interesting finding was that Ancientino (25 μg/mL) did not show an effect of treatment (*p* > 0.05), while Ancientino (12.08 ± 0.94, 100 μg/mL) almost restored the expression of CAT to near those of the control level (11.01 ± 0.78, *p* < 0.01). These findings concur with the in vivo findings mentioned above and further demonstrate the antioxidant effect of Ancientino.

## 4. Discussion

UC has become a worldwide disease threatening human health [18]. Due to its specificity, introducing proper diet and nutritional habits is an essential factor in therapy beyond traditional treatment, but this is often overlooked in medical practice [19]. Despite the fact that there is no special dietary advice for UC, more than 70% of patients point out that inadequate nutrition severely affects the course of UC and aggravates UC severity [20]. Instead, adequate nutrient intake could manage symptoms and slow UC progression [21]. Hence, UC patients urgently seek nutritional supplement therapy to relieve symptoms and improve their quality of life. Ancientino is a complex dietary fiber supplement that mimics the ancient diet by combining grains, legumes, and potato fiber. According to previous studies, Ancientino has been found to possess several physiological activities, such as antioxidant effects, lowering the risk of chronic heart failure, reducing kidney damage, and alleviating symptoms of chronic constipation. However, research on the potential preventative effects of Ancientino on colitis is relatively limited, and the underlying mechanism remains unclear. In the present study, we confirmed dietary supplementation of Ancientino could ameliorate colitis by improving intestinal barrier function, lowering inflammation, and reducing oxidative stress. These results provided valuable scientific evidence supporting the efficacy of Ancientino as a functional food for the adjunctive treatment of UC.

The pathogenesis of UC is marked by an inflammatory response, in which an imbalance between proinflammatory and anti-inflammatory cytokines plays a crucial role [22]. Proinflammatory cytokines can disrupt the normal functioning of the intestinal barrier and lead to increased intestinal permeability [23]. It was observed that IL-1β, IL-6, and TNF-a are overexpressed in IBD patients [24]. Furthermore, TNF-α may promote neutrophilic cell infiltration into colonic tissue, leading to intestinal mucosal microcirculation disorder [25]. Moreover, IL-10 is considered a critical regulator that helps control the duration and severity of colonic inflammation [26]. IL-10 knockout mice could develop colitis spontaneously [27]. The application of TNF-α monoclonal antibodies, such as golimumab, has been shown to effectively relieve the symptoms of UC in patients undergoing treatment [28]. Recent publications have reported that fermented glutinous rice, a dietary supplement, has the potential to mitigate colon inflammation by inhibiting the production of MPO and IL-6 [29]. Zhu et al. observed that gallic acid could effectively restore the diminished levels of IL-10 in mice suffering from colitis induced by DSS, resulting in significant protection against colonic inflammation [30]. The pathology seen in enteritis mice was identical to that seen in human UC, including histologic evidence of weight loss and hemorrhagic diarrhea, epithelial cell death, and neutrophilic infiltration. Furthermore, IL-10 levels were much lower in enteritis mice than in the normal group, while TNF-α, IL-1β, and IL-6 levels were dramatically enhanced after DSS administration. As expected, treatment with different doses of Ancientino significantly downregulated TNF-α, IL-1β, and IL-6 levels and reduced neutrophilic cell infiltration, while significantly elevating IL-10 levels. Moreover, Ancientino was observed to significantly improve colitis symptoms in mice, indicating its potential as an effective anti-inflammatory agent. These data imply that Ancientino was useful in treating colitis by reducing the inflammatory response.

The intestinal barrier loss is seen as another key trigger for the development of UC [31]. An intact barrier serves to prevent the penetration of harmful substances and pathogens into the body’s internal environment, while a compromised barrier allows for their entry, leading to the development of intestinal inflammation [32]. DAO, d-lactate, and endotoxin are often used as chemical markers of intestinal permeability that indirectly reflect gut integrity [33]. DAO is an enzyme that is predominantly located within the intracellular compartments of the intestinal epithelial villi in mammals, while d-lactate and endotoxin are bacterial metabolites produced by the intestinal flora [34]. Once barrier function is impaired and intestinal permeability is abnormally increased, DAO, d-lactate, and endotoxin are released into peripheral blood [35]. In this study, serum levels of DAO, d-lactate, and endotoxin were increased in enteritis mice compared with control mice, indicating that DSS leads to increased intestinal permeability. However, dietary supplementation with Ancientino could oppose the levels of chemical markers above, thereby alleviating intestinal barrier dysfunction.

TJ proteins, which consist of transmembrane proteins such as claudins and occludins, along with zonula occludens (ZO) proteins, play a crucial role in supporting the intestinal barrier. These proteins bridge the gap of both adjacent cells and stabilize the intestinal barrier [36]. Studies have indicated that a decrease in TJ protein expression can compromise adherence junction integrity, ultimately resulting in increased gut permeability and disruption of immune homeostasis [37]. Abnormal TJ protein expression has been identified in UC patients, which has been associated with increased intestinal permeability [38]. In addition, the deficiency of TJs was reported to result in colitis and colitis-associated intestinal carcinoma progression [39]. Conversely, increasing TJ protein expression has been found to maintain the intestinal epithelial barrier integrity and inhibit colon inflammation [40]. The evidence discussed above suggests that the upregulation of TJ protein expression may represent a viable approach for restoring proper colon function and combating UC. The findings of this study indicated that the expression of TJs (ZO-1 and occludin) was significantly reduced in enteritis mice, leading to a disruption in intestinal barrier function. However, treatment with Ancientino resulted in a dose-dependent increase in the levels of the two TJs above, effectively restoring the integrity of the intestinal barrier. Consistently, the IF analysis exhibited similar results. According to all of the aforementioned findings, Ancientino reduced the degree of intestinal inflammation by suppressing not merely the infiltration of inflammatory cells yet additionally the disruption of the intestinal barrier by modulating the expression of TJs.

Oxidative stress is believed to play a crucial role in the initiation of UC [41]. During UC, macrophages and neutrophils generate high levels of ROS, such as O^2−^ and H_2_O_2_. The resulting oxidative damage can lead to membrane disruption and eventual injury to colon tissue [42]. However, the body’s endogenous antioxidants, such as SOD and CAT, are capable of counteracting the effects of free radicals. Additionally, GSH-PX works in conjunction with SOD to establish a comprehensive defense mechanism that can protect the colon mucosa from peroxide-induced damage [43,44]. MDA is a significant intermediate product of the oxidation of membrane lipids. It could exacerbate existing membrane injury, resulting in cytotoxicity, inflammatory response, and necrosis [45,46]. Nowadays, MDA levels are widely used as a means of evaluating the degree of tissue damage, the strength of lipid peroxidation, and the concentration of oxygen free radicals in various biological systems [47]. Zhang et al. demonstrated that supplementation with *Rhus chinensis* Mill. fruits can prevent oxidative stress response, such as a decreased MDA level and increased total GSH and SOD activity, thereby inhibiting colitis [48]. Furthermore, *L. plantarum* JS19 relieved DSS-induced colitis by decreasing lipid peroxidation and increasing antioxidant enzyme activity [49]. It follows that enhancing antioxidant capacity is one of the important strategies for UC therapy. Consistent with these findings, this study showed that DSS promoted oxidative stress, as evidenced by decreased SOD, CAT, and GSH-PX activities, as well as a high amount of MDA in the colon tissue of DSS-induced enteritis animals and LPS-stimulated Caco-2 cells, and Ancientino dramatically reversed these changes. The change was more pronounced in the group receiving 1200 mg/kg of Ancientino. The findings above show that oxidative stress is linked to the advancement of UC. Ancientino can eliminate excess free radicals and increase oxidation resistance. It also reduces the inflammatory response, minimizing colon tissue damage, which may be one of Ancientino’s modes of avoiding UC.

Due to its distinct function, dietary fiber is referred to as the “seventh nutrient”. Dietary fiber cannot be digested or absorbed in the body [50]. Still, intestinal flora can use it as a fermentation substrate to produce beneficial substances such as short-chain fatty acids (SCFAs) [51]. These substances impact intestinal health by modifying the gut environment and permeability. Preclinical animal models have widely demonstrated the positive results of fiber supplements. Fermented barley and soybean were utilized as dietary fiber substitutes in enteritis mice. The study demonstrated that a barley–soybean combination elevated tight junction protein levels, reduced FITC-dextran permeability, and maintained the intestinal barrier’s integrity [52]. Another study discovered that the mucosal pathogen *C. rodentium* induced fatal colitis in fiber-free mice, while fiber consumption improved colonic inflammation and weight loss in fiber-rich animals [53]. In the DSS-induced colitis-associated colon carcinogenesis (CAC) model, Nishiguchi et al. found that a high-fructose diet exacerbated chronic colitis. However, they also found that a fiber-rich diet protected mice from chronic colitis tumorigenesis [54]. In clinical trials, dietary fiber supplementation has shown promising results in patients with UC. A pilot research study looked at how TNF-α, IL-6, and IL-8 serum levels in UC patients with remission were affected by germinated barley foodstuff (GBF). Patients in the treatment group received normal medication therapy and 30 g of GBF daily via oral administration during the two-month study. Throughout the research, the above inflammatory factors levels fell in the GBF group relative to controls [55]. Another crossover trial involving 17 UC patients who were in remission or mild found similar results. They randomly allocated patients to one of two diets: a low-fat, high-fiber diet or a regular American diet. The results show a low-fat, high-fiber diet suppressed inflammation response [56]. Therefore, increasing dietary fiber intake is necessary to help maintain intestinal health. Furthermore, dietary fiber has been found to provide other health advantages, such as lowering the risk of chronic illnesses such as diabetes, heart disease, and some cancers [57].

Overall, the literature supports the potential benefits of dietary fiber in human nutrition and health. While specific recommendations for fiber intake may vary depending on individual needs and health goals. Adults are typically advised to ingest at least 20–35 g of fiber per day from a range of sources [58,59], such as grains, fruits, and vegetables. However, because the fiber content of fresh fruits and vegetables is insufficient to meet this assumption, Ancientino might be utilized as a supplemental source of dietary fibers in the human diet. In this study, mice received Ancientino (1200 mg/kg/day) at a dose equivalent to ~10 g of dietary fiber daily in humans (calculation based on the ratio of mouse-to-human surface area). Ancientino has up to 10 g of dietary fiber in each packet. Therefore, two packets per day can adequately compensate for a shortage of dietary fiber.

In this study, although we carried out multifaceted research on Ancientino, the exact mechanisms of the Ancientino effect on UC still need to be thoroughly explored. Moreover, it remains uncertain if it applies to a chronic enteritis model, and whether Ancientino can affect the composition of the intestinal flora also needs further research. Based on the data presented in this study and findings from the literature [60,61], it is speculated that the mechanism of action of Ancientino is likely mediated by various signaling pathways, including the AMP-activated protein kinase (AMPK) pathway, the nuclear factor-kappa B (NF-κB) pathway, and the Wnt pathway. For example, it is possible that Ancientino may act as a prebiotic, promoting the growth of beneficial gut bacteria and producing SCFAs. These SCFAs have the ability to activate the AMPK pathway, which controls energy metabolism and inflammation. Additionally, SCFAs can inhibit the NF-κB pathway, reducing inflammation and oxidative stress. Furthermore, SCFAs can activate the Wnt pathway, promoting cell proliferation and differentiation. These signaling pathways may contribute to the beneficial effects of Ancientino on gut and overall health. In the follow-up experiment, we will conduct relevant experiments to explore its in-depth mechanism, which provides a more comprehensive supplement for Ancientino’s nutritional supplement therapy.

## 5. Conclusions

In summary, this study demonstrated that Ancientino alleviated DSS-induced colitis and exerted an anticolitis effect by reducing inflammatory response, suppressing oxidative stress, and repairing colonic barrier function. Thus, Ancientino might be an effective therapeutic dietary resource for UC.

## Figures and Tables

**Figure 1 nutrients-15-02798-f001:**
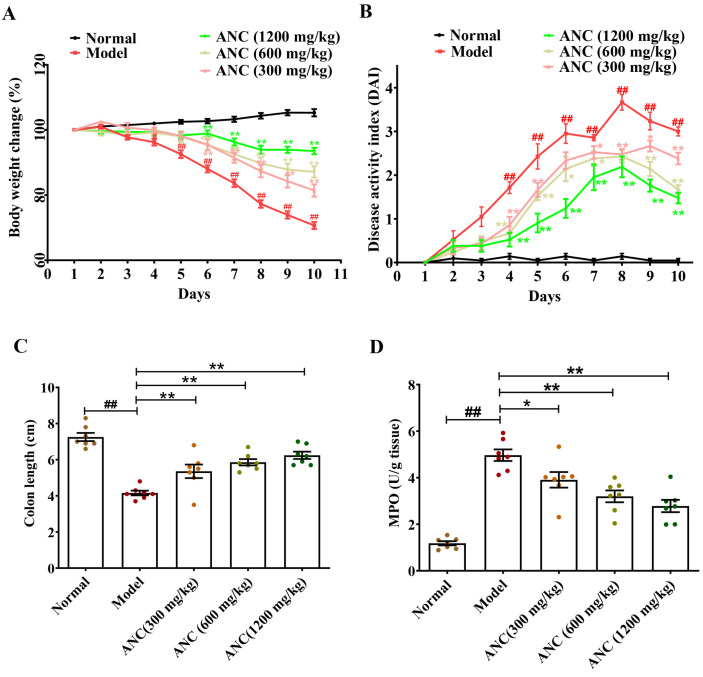
Ancientino (ANC) alleviated the symptoms of enteritis mice: (**A**) Weight loss. (**B**) DAI. (**C**) Colon length. (**D**) MPO. (**E**,**F**) H&E staining and histopathologic scores. Every dot in the figure represents a mouse, and different colors represent different groups. Data are mean ± SEM of 7 mice per group. ^#^
*p* < 0.05, ^##^
*p* < 0.01 vs. the normal group, * *p* < 0.05, ** *p* < 0.01 vs. the model group.

**Figure 2 nutrients-15-02798-f002:**
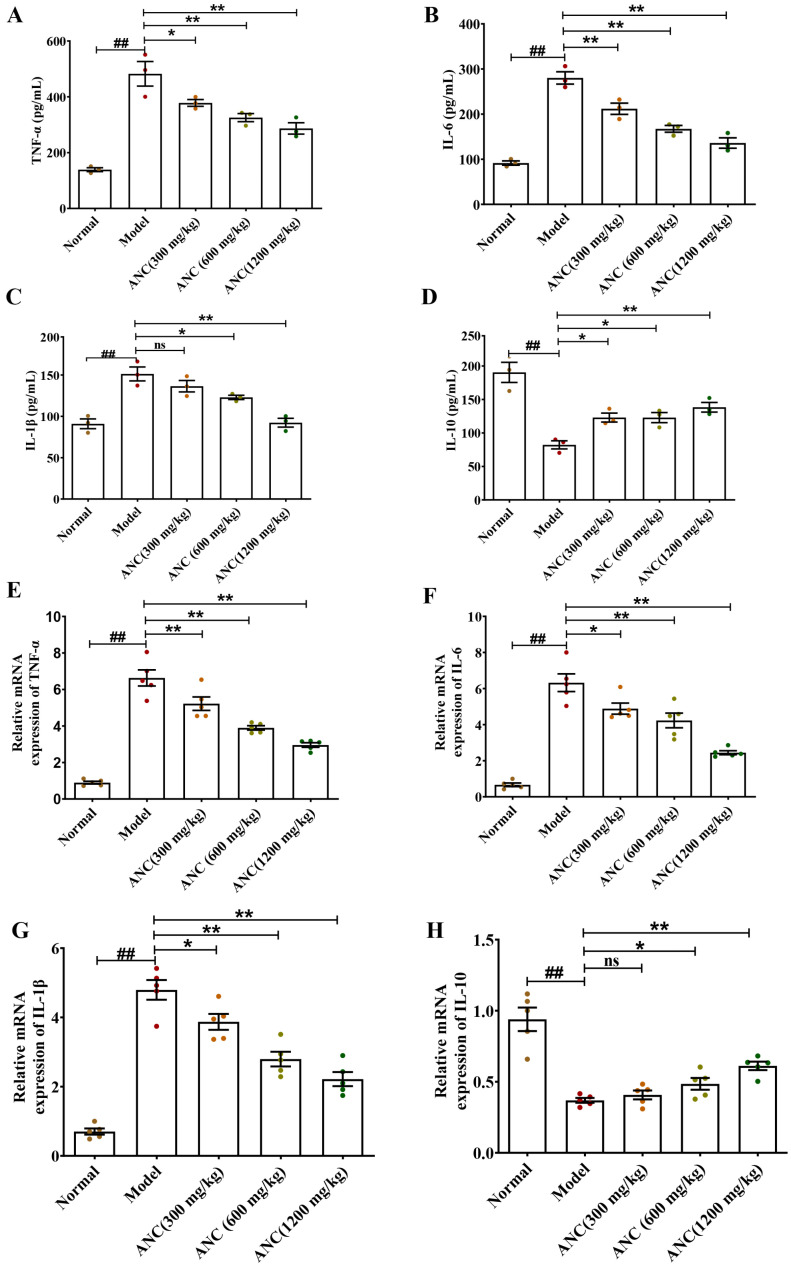
The impact of Ancientino (ANC) on the levels of inflammatory cytokines in enteritis mice: TNF-α (**A**), IL-6 (**B**), IL-1β (**C**), and IL-10 (**D**) protein levels in colonic homogenates were determined using ELISA. qRT-PCR was used to quantify the TNF-α (**E**), IL-6 (**F**), IL-1β (**G**), and IL-10 (**H**) mRNA levels in colonic homogenates. Every dot in the figure represents a mouse, and different colors represent different groups. The data are presented as the mean ± SEM of 3 (ELISA) and 5 (qPCR) mice for every group, respectively. ^##^
*p* < 0.01 vs. the normal group; ns indicates *p* > 0.05, * *p* < 0.05, ** *p* < 0.01 vs. the model group.

**Figure 3 nutrients-15-02798-f003:**
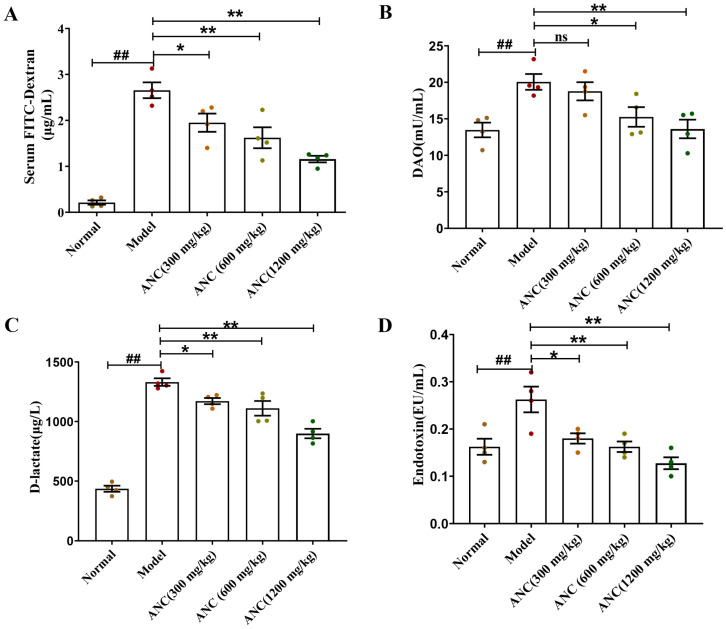
Effect of Ancientino (ANC) on the intestinal permeability in enteritis mice: (**A**) FITC-Dextran. (**B**) DAO. (**C**) D-lactate. (**D**) endotoxin. Every dot in the figure represents a mouse, and different colors represent different groups. Data are mean ± SEM of 4 mice per group. ^##^
*p* < 0.01 vs. the normal group; ns indicates *p* > 0.05, * *p* < 0.05, ** *p* < 0.01 vs. the model group.

**Figure 4 nutrients-15-02798-f004:**
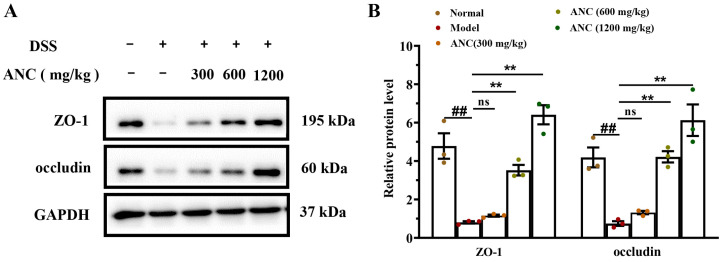
Effect of Ancientino (ANC) on the expression of TJs in enteritis mice: (**A**,**B**) WB of ZO-1 and occludin and statistical analysis of WB in the colonic tissue. (**C**,**D**) qRT-PCR was used to quantify the ZO-1 and occludin mRNA levels in colonic tissue. Every dot in the figure represents a mouse, and different colors represent different groups. The data are presented as the mean ± SEM of 3 (ELISA) and 5 (qPCR) mice per group, respectively. ^##^
*p* < 0.01 vs. the normal group; ns indicates *p* > 0.05, ** *p* < 0.01 vs. the model group.

**Figure 5 nutrients-15-02798-f005:**
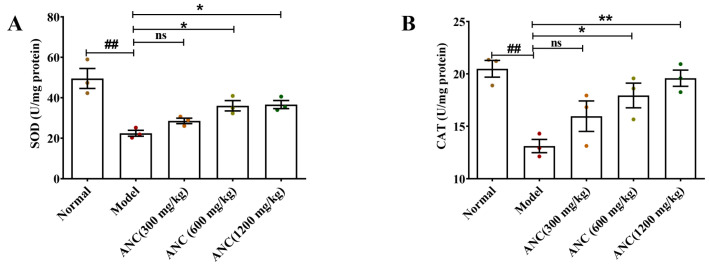
The effects of Ancientino (ANC) on the levels of oxidative stress biomarkers in vivo: The level of SOD (**A**), CAT (**B**), MDA (**C**), and GSH-Px (**D**) in enteritis mice. Every dot in the figure represents a mouse, and different colors represent different groups. Data are mean ± SEM of 3 mice per group. ^##^
*p* < 0.01 vs. the normal group; ns indicates *p* > 0.05, * *p* < 0.05, ** *p* < 0.01 vs. the model group.

**Figure 6 nutrients-15-02798-f006:**
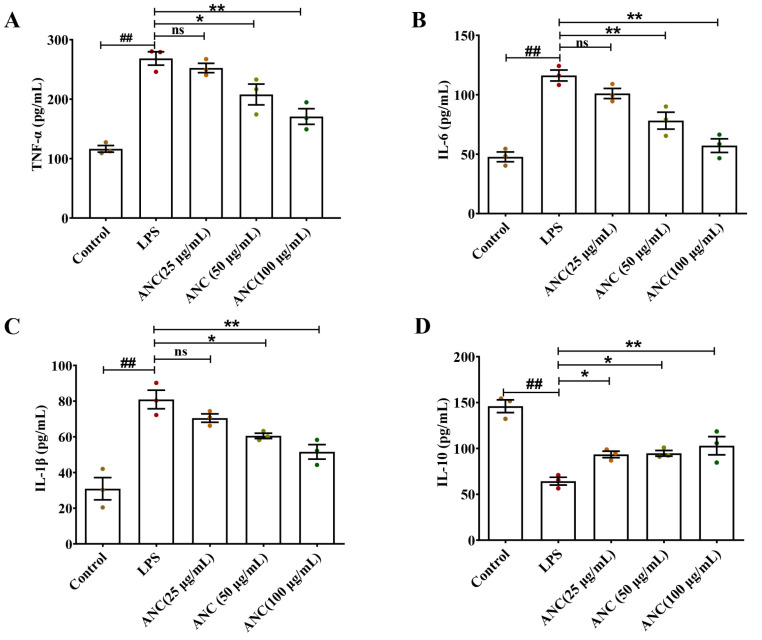
Effect of Ancientino (ANC) on LPS (1 μg/mL)-induced Caco-2 cells’ secretion of inflammatory cytokines: The levels of TNF-α (**A**), IL-6 (**B**), IL-1β (**C**), and IL-10 (**D**) protein expression were determined using ELISA. The mRNA expression levels of TNF-α (**E**), IL-6 (**F**), IL-1β (**G**), and IL-10 (**H**) were assessed by qRT-PCR. Every dot in the figure represents an independent experiment, and different colors represent different groups. The results are expressed as the mean ± SEM, with data obtained from 3 independent experiments. ^##^
*p* < 0.01 vs. the control group; ns indicates *p* > 0.05, * *p* < 0.05, ** *p* < 0.01 vs. the LPS (1 μg/mL) group.

**Figure 7 nutrients-15-02798-f007:**
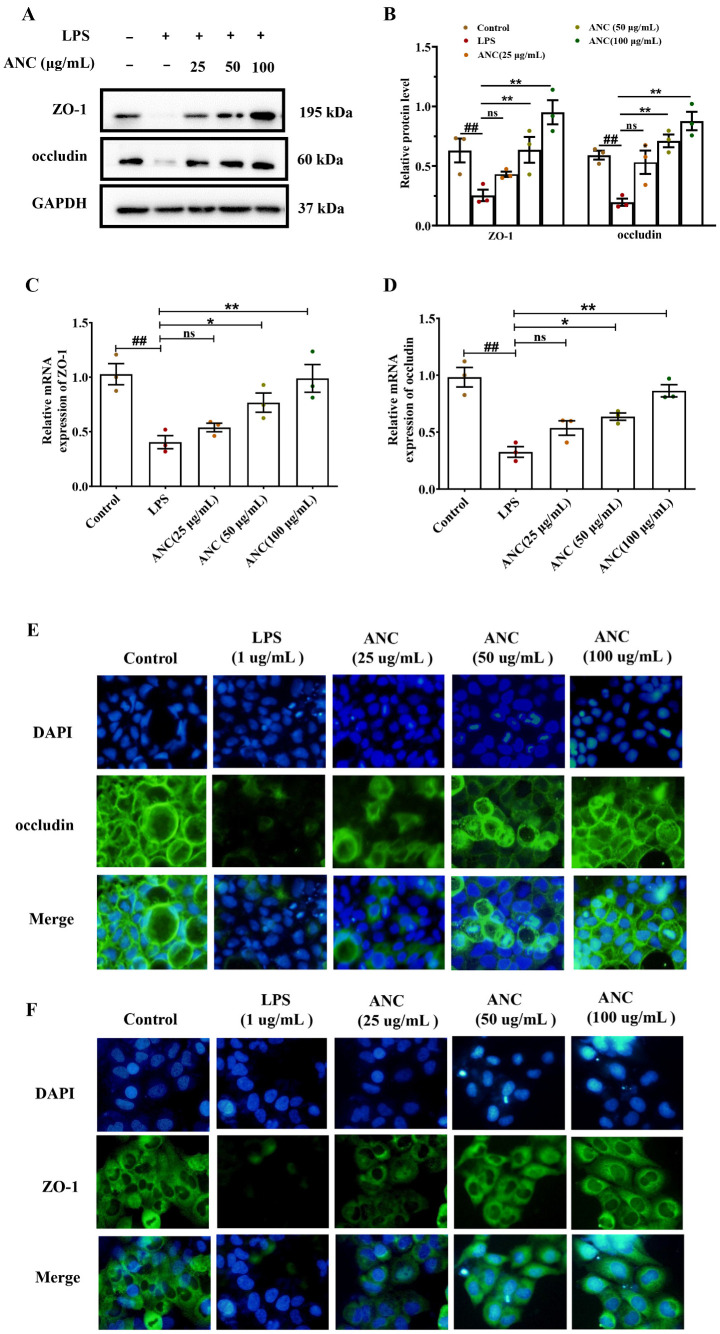
Effect of Ancientino (ANC) on the expression of TJs in LPS (1 μg/mL)-induced Caco-2 cells by WB, qRT-PCR, and IF: (**A**) WB of ZO-1 and occludin and (**B**) statistical analysis of WB. (**C**,**D**) qRT-PCR analysis of ZO-1 and occludin. (**E**,**F**) ZO-1 and occludin levels based on IF staining (200×). Every dot in the figure represents an independent experiment, and different colors represent different groups. Data are presented as the mean ± SEM, with data obtained from 3 independent experiments. ^##^
*p* < 0.01 vs. the control group; ns indicates *p* > 0.05, * *p* < 0.05, ** *p* < 0.01 vs. the LPS (1 μg/mL) group.

**Figure 8 nutrients-15-02798-f008:**
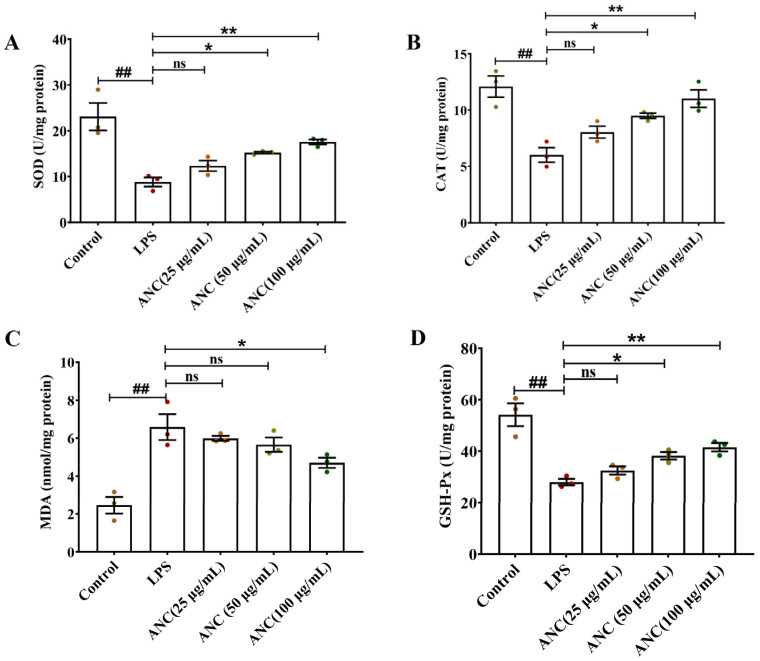
The effects of Ancientino (ANC) on the levels of oxidative stress biomarkers in Caco-2 cells: The level of (**A**) SOD, (**B**) CAT, (**C**) MDA, and (**D**) GSH-Px. Every dot in the figure represents an independent experiment, and different colors represent different groups. Data are presented as mean ± SEM, with data obtained from 3 independent experiments. ^##^
*p* < 0.01 vs. the control group; ns indicates *p* > 0.05, * *p* < 0.05, ** *p* < 0.01 vs. the LPS (1 μg/mL) group.

**Table 1 nutrients-15-02798-t001:** The DAI score criteria.

DAI Score	Weight Loss (%)	Stool Consistency	Fecal Occult Blood
0	0	Normal	Normal
1	1–5	Loose stools (mild)	Hemoccult positive
2	5–10	Loose stools (medium)	
3	10–20	Loose stools (high)	
4	>20	Diarrhea	Gross bleeding

**Table 2 nutrients-15-02798-t002:** Histological score criteria.

Score	Inflammation Severity	Inflammatory Locations	Crypt Damage
0	none	none	none
1	slight	mucosa	basal third damaged
2	moderate	mucosa and submucosa	basal two-thirds damaged
3	severe	transmural	only surface intact
4			entire crypt and epithelium lost

**Table 3 nutrients-15-02798-t003:** Primer sequences.

Name	Sequence (5′ to 3′)	GenBank Accession Number
TNF-α	F-TGTCCCTTTCACTCACTGGC	NM_013693.3
	R-TCTTCTGCCAGTTCCACGTC	
IL-1β	F-TCAGCACCTCACAAGCAGAG	NM_008361.4
	R-TTCTTGTGACCCTGAGCGAC	
IL-6	F-TTGGGACTGATGCTGGTGAC	NM_031168.2
	R-AGACAGGTCTGTTGGGAGTG	
IL-10	F-TGAATTCCCTGGGTGAGAAG	M84340.1
	R-TTTGTTGGGTGGCTCTAAGG	
occludin	F-TTGACTGGGCTGAACACTCC	NM_008756.2
	R-ACATCACAGCTCACACCAGG	
ZO-1	F-AAACAGCCCTACCAACCTCG	BC138028.1
	R-TTCGAGGCAGCTGCTCATAG	
GAPDH	F-CAGCAAGGACACTGAGCAAG	NM_001289726.1
	R-GGTCTGGGATGGAAATTGTG	

## Data Availability

Not applicable.

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
