# Peer review of "Dietary Supplementation of Ancientino Ameliorates Dextran Sodium Sulfate-Induced Colitis by Improving Intestinal Barrier Function and Reducing Inflammation and Oxidative Stress"

_nutrients, 2023, doi:10.3390/nu15122798_

Round 1

Reviewer 1 Report

Dear authors, I appreciate the work done. The article brings several results from animal data and cell culture. Below are some suggestions to improve the presentation of the article.

- It is not clear whether the authors have a conflict of interest with Guilin Ancientino Food Technology Co., Ltd". Please explain.

- How was the treatment dose of Ancientino estimated? How does this dose represent consumption for humans? This issue needs to be discussed in the discussion section.

- The authors did not describe the ethics committee approval number for the use of experimental animals.

- What is “tool bleeding”? I believe it was a typo for stool bleeding. There are more typos throughout the text; please check it.

- I suggest that the authors add the lot number of the dosage kits so that the reader knows where to look for the protocol/datasheet.

- Histological analysis and histological scoring must be described in more detail.

- How does the intestinal permeability analysis was calculated? This must be described in more detail.

- The authors describe that "After a 6-hour starvation period, the cells were co-treated with each drug and LPS (1 μg/mL) for 24 hours." Which drugs? These details need to be described in the methodology.

- How was the treatment concentration of Ancientino estimated (in vitro studies)?

- Did the authors not perform Ancientino toxicity testing for caco-2 cells before starting the experiments (e.g. MTT, PrestoBlue)?

- Were measurements of the levels of MDA, CAT, SOD, and GSH-Px made in the supernatant or cell lysate?

- Were the data tested for normality before statistical analysis?

- The data has not been fully described. I suggest that authors describe the data, not just point out the p-value. That is, describe the means or median, error or deviation, differences in %, etc.

- I believe that the results would be better exposed if the animal data were displayed first and then the cell data.

- The authors mentioned at the end of the discussion that more experiments will be carried out to unravel Ancientino's mechanism of action. However, I believe that the authors could speculate the mechanism of action of Ancientine with the data found here and with data from the literature. Still, I would like the authors to talk about how this supplement could act in cell culture, since in this case, there is no presence of gastric mucus, microbiota and several other factors that are found in a live organism.

There are some typos that need to be corrected.

Author Response

Dear authors, I appreciate the work done. The article brings several results from animal data and cell culture. Below are some suggestions to improve the presentation of the article.

  1. It is not clear whether the authors have a conflict of interest with Guilin Ancientino Food Technology Co., Ltd". Please explain.

Response: Thank you for your attention. There is no conflict of interest with Guilin Ancientino Food Technology Co., Ltd.

  1. How was the treatment dose of Ancientino estimated? How does this dose represent consumption for humans? This issue needs to be discussed in the discussion section.

Response: Thank you for your attention. As suggested, we have added this content in the discussion section of the latest manuscript (see lines 508-515).

Adults are typically advised to ingest at least 20-35 grams of fiber per day from a range of sources [1,2]. such as grains, fruits and vegetables. However, because the fiber content of fresh fruits and vegetables is insufficient to meet this assumption, Ancientino might be utilized as a supplemental source of dietary fibers in the human diet. In this study, Mice receiving Ancientino (1200 mg/kg/day) is equivalent to ~10 g of dietary fiber daily in humans (calculation based on the ratio of mouse to human surface area). Ancientino has up to 10 g of dietary fiber in each packet. Therefore, two packets per day can adequately compensate for a shortage of dietary fiber.

Assuming that the clinical dose of human (70 kg) is X mg/kg, the dose of mouse (20 g) is:

X mg/kg×70 kg×0.0026/20 g=X mg/kg×70kg×0.0026/0.02kg= 9.1 X mg/kg.

The dose of Ancientino: 20 g/70 kg×9.1=2570 mg/kg, the maximum dose was halved to 1200 mg/kg because the dose (2570 mg/kg) was too dense for intragastrically administered mice.

  1. The authors did not describe the ethics committee approval number for the use of experimental animals.

Response: Thank you for your attention. In the initial manuscript, we have described the ethics committee approval number for the use of experimental animals (see lines 548-549), “This study was approved by the Animal Ethics Committee of Guilin Medical University, and the animal protocol approval number is GLMC202103003.”

  1. What is “tool bleeding”? I believe it was a typo for stool bleeding. There are more typos throughout the text; please check it.

Response: We sincerely thank the reviewer for careful reading. We are sorry for our carelessness. As suggested by the reviewer, we have corrected the “tool bleeding” into “stool bleeding”. (see line 113).

  1. I suggest that the authors add the lot number of the dosage kits so that the reader knows where to look for the protocol/datasheet.

Response: Thank you for your attention. As suggested, we have added the lot number of the dosage kits (see lines 67-84).

DSS (S5036) of MW 36-50 kDa was from MPBio (Solon, USA). Fluorescein isothi-ocyanate (FITC)-dextran (60842-46-8) of MW 3000–5000 kDa and Lipopolysaccharide (LPS, L3129) were from Sigma-Aldrich (St Louis, MO). Tumor necrosis factor-alpha (TNF-α, CT303A), interleukin-10 (IL-10, CT307A), interleukin 6 (IL-6, CT299A) and interleukin-1 beta (IL-1β, 1210122) enzyme-linked immunosorbent assay (ELISA) kits were supplied by Dakowei Biotech Co., Ltd. (Beijing, China). Myeloperoxidase (MPO, A044-1-1), Malondialdehyde (MDA, A003-1), Catalase (CAT, A007-2), Superoxide dismutase (SOD, A001-3), and Glutathione peroxidase (GSH-Px, A005) assay kits were from Nanjing Jiancheng Bioengineering Company (Nanjing, China). The concentra-tions of serum Diamine Oxidase (DAO, CSB-E10090m) and endotoxin (CSB-E13066m)ELISA kits were from Cusabio LLC; D-lactate (JL48176)ELISA Kit was obtained from Shanghai Future Industrial Limited by Share Ltd. (Shanghai, China); FastKing gDNA Dispelling RT Super Mix (KR118) and SYBR Green (FP205) were from TIANGEN Co., Ltd. (Beijing, China). Main primary antibodies: Rabbit monoclonal an-tibodies recognizing ZO-1 (AF5145) and occludin (DF7504) were provided by Affinity Biosciences (Minnesota, USA). Dilutions used were 1:1000 for western blots and 1:200 for immunofluorescence experiments. Mouse monoclonal antibodies recognizing GAPDH (1:1000, TA-08) was from Beijing Zhongshan Golden Bridge Biotechnology Co., Ltd. Goat-anti-rabbit (A21020) and goat-anti-mouse (A21010) immunoglobulin G secondary antibody (1:10000) was from Abbikine, Inc. (Redlands, CA, USA). All chemical reagents were of analytical grade or above.

  1. Histological analysis and histological scoring must be described in more detail.

Response: Thank you for your attention. First, I'm sorry for not specifically describing Histological analysis and histological scoring in the method. We have described it in detail in the latest manuscript. (see lines 127-130).

The scores were graded as follows: a) severity of inflammation: 0 = none; 1 =mild; 2 = mod erate;3 = severe; b) inflammatory locations: 0 = none; 1 = mucosa; 2 = mucosa and submucosa; 3 = transmural; c) crypt damage: 0 =none; 1 = basal 1/3 damaged; 2 = basal 2/3 damaged; 3 =only surface epithelium intact; 4 = entire crypt and epithelium lost. The histological score was determined by the summation of three evaluations, resulting in a maximum score of 10.

  1. How does the intestinal permeability analysis was calculated? This must be described in more detail.

Response: Thank you for your attention. First, I'm sorry for not specifically describing intestinal permeability analysis in the method. We have described it in detail in the latest manuscript. (see lines 133-140).

Serum FITC-dextran concentration was used to evaluate intestinal permeability in vivo. Briefly, after being fed with DSS for seven days, the mice were fasted for four hours and then orally administered 0.3 mL FITC-Dextran (6 mg/10 g). After three hours, bloods were collected from each mouse from the orbital sinuses. The blood was placed into 1.5 mL Eppendorf tubes and centrifuged at 3500 rpm for 10 min at 4°C. Then 0.1 mL serum samples were measured using a Synergy HT plate reader (BioTek, USA). The fluorescence intensity of the serum (excitation at 485 nm; emission at 535 nm) was used to determine the level of intestinal permeability. Serum FITC-dextran concentrations were calculated from a standard curve of serially diluted FITC-dextran in PBS.

  1. The authors describe that "After a 6-hour starvation period, the cells were co-treated with each drug and LPS (1 μg/mL) for 24 hours." Which drugs? These details need to be described in the methodology.

Response: Thank you for your attention. “each drug” in the manuscript refers to a different dose of Ancientino. For more accurate descriptions, we have changed “each drug” to “different doses of Ancientino (25, 50, 100 μg/mL)” (see lines 151-152).

  1. How was the treatment concentration of Ancientino estimated (in vitro studies)?

Response: Thank you for your attention. We performed Ancientino toxicity testing for caco-2 cells before starting the experiments by CCK8 assay, Experimental results are displayed in the supplementary materials of the initial manuscript (see supplementary materials in the attached word).

Cell viability

Relative cell viability was measured using the cell counting kit 8 (CCK-8) assay. Caco-2 cells were dispensed in 96-well plates at 7×103 cells per well. After cell adherence, cells were administered with different dose of Ancientino (0, 12.5, 25, 50, 100 μg/mL) for 24 h. Then CCK-8 solution was added to each well and incubated at 37℃ for 4 h. The absorbance was measured at 450 nm to calculate the relative cell viability.

To investigate the anti-UC activity of Ancientino in vitro, Caco-2 cells were selected for research. The cells were treated for 24 h with different dose of Ancientino (0, 12.5, 25, 50,100 μg/mL) and the relative viability of the cells was measured. As shown below the graph, Ancientino-treated groups exhibited no significant difference on the relative cell viability compared to the control group. The results demonstrate that Ancientino has no apparent cytotoxicity to the Caco-2 cells. Based on these preliminary data 25, 50,100 μg/mL were selected in the succedent experiments.

  1. Did the authors not perform Ancientino toxicity testing for caco-2 cells before starting the experiments (e.g. MTT, PrestoBlue)?

Response: Thank you for your attention. We performed Ancientino toxicity testing for caco-2 cells before starting the experiments by CCK8 assay, Experimental results are displayed are detailed above (see question 9).

  1. Were measurements of the levels of MDA, CAT, SOD, and GSH-Px made in the supernatant or cell lysate?

Response: Thank you for your attention. Colon tissues were washed with ice-cold PBS and homogenized, then the samples were subjected to centrifugation at 3500 rpm for 10 minutes at 4 °C to obtain the colonic supernatant. For Caco-2 cells, the cell supernatant after LPS stimulation for 24 h was collected, and the levels of MDA, CAT, SOD, and GSH-Px were determined by using a detection kit.

  1. Were the data tested for normality before statistical analysis?

Response: Thank you for your attention. The normality (Shapiro-Wilk) test was performed before the statistical analysis. Only data that passed the normality test (p ˃ 0.05) were considered for statistical analyses.

  1. The data has not been fully described. I suggest that authors describe the data, not just point out the p-value. That is, describe the means or median, error or deviation, differences in %, etc.

Response: We think this is an excellent suggestion. Without changing the smoothness of the article, we added the description of some results (means or median, differences in %, etc.) and displayed the original results on Figure to make the article expression clearer (see results section for details).

  1. I believe that the results would be better exposed if the animal data were displayed first and then the cell data.

Response: Thank you for your attention, we are willing to defer to the reviewer’s view. We have displayed the animal data first and then the cell data in the latest manuscript (see results section for details).

  1. The authors mentioned at the end of the discussion that more experiments will be carried out to unravel Ancientino's mechanism of action. However, I believe that the authors could speculate the mechanism of action of Ancientine with the data found here and with data from the literature. Still, I would like the authors to talk about how this supplement could act in cell culture, since in this case, there is no presence of gastric mucus, microbiota and several other factors that are found in a live organism.

Response: Thank you for your attention. We have supplemented this content in the discussion (see lines 519-529).

Based on the data presented in this study and findings from the literature[3,4], it is speculated that the mechanism of action of Ancientino is likely mediated by various signaling pathways, including the AMP-activated protein kinase (AMPK) pathway, the nuclear factor-kappa B (NF-κB) pathway, and the Wnt pathway. For example, it is possible that Ancientino may act as a prebiotic, promoting the growth of beneficial gut bacteria and producing short-chain fatty acids (SCFAs). These SCFAs can activate the AMPK pathway, which regulates energy metabolism and inflammation. Additionally, SCFAs can inhibit the NF-κB pathway, reducing inflammation and oxidative stress. Furthermore, SCFAs can activate the Wnt pathway, promoting cell proliferation and differentiation. These signaling pathways may contribute to the beneficial effects of Ancientino on gut and overall health.

Regarding the effects of Ancientino in cell culture, it is important to note that the absence of gastric mucus, microbiota, and other factors found in a live organism can limit the interpretation of results. However, cell culture studies can provide valuable insights into the potential mechanisms of action of supplements such as Ancientino. In vitro cell experiments, Ancientino can affect the growth, metabolism and behavior of cells in various ways. First of all, Ancientino can be used as a nutrition source of cells, providing energy and nutrients required for cells. Adding Ancientino to cell culture medium can promote the growth and proliferation of cells, and can also promote the differentiation and functional expression of cells. Secondly, Ancientino can affect the metabolism and signal transformation of cells. Ancientino can be decomposed by enzymes in the cells into simple sugars, providing cell energy needs. In addition, Ancientino can also affect the metabolic pathway of cells, such as the glycolytic and the fatty acid oxidation pathways, etc. Ancientino can also affect the signal transition pathway of cells, such as AMPK, NF-κB, WNT and other pathways, thereby regulating the growth, differentiation, apoptosis and other processes of cells.

Reviewer 2 Report

This paper describes that the oral administration of “Ancientino“ to rats and its administration in vitro to Caco-2 cells has beneficial effects in terms of inflammation regulation. As “Ancientino“ ius a combination of fruit components including fibers that is not very surprising. Although the number of repetitions of the experiments are quite low, the study seems to be well performed, there are some minor flaws and  the overall importance of that study is not overwhelming.

·         The Material and Method section need to be improved. Here we need to know what exactely has been done and how. For example is to add which antibodies (primary and secondary) have been used (clone, source), how many experiments have been performed, describing technical and biological replication. It was stated that three experiment have been performed. Doers it means tree wells on a plate, does is means tree different plates on different days and so on. It has to be described in a way that the experiments can be repeated in a different laboratory.  

·         It is somehow ridiculous to perform anova with post hock test which requires homogeneous variances when the number of experiments in 3. I recommend to show the three measurements directly.  

·         The exact numbers of repetitions of an experiment should be stated in the resp. figures

·         As far as I understand were all the experiments performed with one batch of Ancientino. It should be possible to include the exact recipe in the paper, and no estimations. Analysis of the Ancientino should be performed and added to the publication  

·         Relevant Literature concerning fibers in Nutrition of humans including clinical trials have not been cited. It is quite common, that oral administration of fiber has beneficial effects on UC, therefore that should be discussed properly.

There are only a few typing errors.

Author Response

This paper describes that the oral administration of “Ancientino” to rats and its administration in vitro to Caco-2 cells has beneficial effects in terms of inflammation regulation. As “Ancientino” is a combination of fruit components including fibers that is not very surprising. Although the number of repetitions of the experiments are quite low, the study seems to be well performed, there are some minor flaws and the overall importance of that study is not overwhelming.

  1. The Material and Method section need to be improved. Here we need to know what exactely has been done and how. For example: is to add which antibodies (primary and secondary) have been used (clone, source), how many experiments have been performed, describing technical and biological replication. It was stated that three experiment have been performed. Does it mean three wells on a plate, does is means three different plates on different days and so on. It has to be described in a way that the experiments can be repeated in a different laboratory.

Response: We feel great thanks for your professional review work on our article. As you are concerned, there are several problems that need to be addressed. According to your nice suggestions, we have made extensive corrections to our previous draft, the details are added in the latest manuscript. As suggested, we have added the lot number of the dosage kits (see lines 67-84) and described antibodies (primary and secondary) have been used (clone, source). Three experiment is means three different plates on different days. In addition, we have added the specific biology repetition in all legends.

DSS (S5036) of MW 36-50 kDa was from MPBio (Solon, USA). Fluorescein isothi-ocyanate (FITC)-dextran (60842-46-8) of MW 3000–5000 kDa and Lipopolysaccharide (LPS, L3129) were from Sigma-Aldrich (St Louis, MO). Tumor necrosis factor-alpha (TNF-α, CT303A), interleukin-10 (IL-10, CT307A), interleukin 6 (IL-6, CT299A) and interleukin-1 beta (IL-1β, 1210122) enzyme-linked immunosorbent assay (ELISA) kits were supplied by Dakowei Biotech Co., Ltd. (Beijing, China). Myeloperoxidase (MPO, A044-1-1), Malondialdehyde (MDA, A003-1), Catalase (CAT, A007-2), Superoxide dismutase (SOD, A001-3), and Glutathione peroxidase (GSH-Px, A005) assay kits were from Nanjing Jiancheng Bioengineering Company (Nanjing, China). The concentra-tions of serum Diamine Oxidase (DAO, CSB-E10090m) and endotoxin (CSB-E13066m)ELISA kits were from Cusabio LLC; D-lactate (JL48176)ELISA Kit was obtained from Shanghai Future Industrial Limited by Share Ltd. (Shanghai, China); FastKing gDNA Dispelling RT Super Mix (KR118) and SYBR Green (FP205) were from TIANGEN Co., Ltd. (Beijing, China). Main primary antibodies: Rabbit monoclonal an-tibodies recognizing ZO-1 (AF5145) and occludin (DF7504) were provided by Affinity Biosciences (Minnesota, USA). Dilutions used were 1:1000 for western blots and 1:200 for immunofluorescence experiments. Mouse monoclonal antibodies recognizing GAPDH (1:1000, TA-08) was from Beijing Zhongshan Golden Bridge Biotechnology Co., Ltd. Goat-anti-rabbit (A21020) and goat-anti-mouse (A21010) immunoglobulin G secondary antibody (1:10000) was from Abbikine, Inc. (Redlands, CA, USA). All chemical reagents were of analytical grade or above.

  1. It is somehow ridiculous to perform anova with post hock test which requires homogeneous variances when the number of experiments in 3. I recommend to show the three measurements directly.

Response: We think this is an excellent suggestion. Without changing the smoothness of the article, we added the description of some results and displayed the original results on figures to make the article expression clearer (see results section for details).

  1. The exact numbers of repetitions of an experiment should be stated in the resp. figures

Response: Thank you for your attention. The exact number of repetitions of an experiment has been stated in the resp. figures (see legends of results section for details).

  1. As far as I understand were all the experiments performed with one batch of Ancientino. It should be possible to include the exact recipe in the paper, and no estimations. Analysis of the Ancientino should be performed and added to the publication.

Response: Thank you for your attention. I'm sorry for not specifically describing the exact recipe of Ancientino. After communication with the partner (Guilin Ancientino Food Technology Co., Ltd), we learned that the specific formula of Ancientino used in the experiment is as follows:

Composite dietary fiber is composed of the following components in proportion to the weight percentage w/w:

  • The ratio of xylan and glucomannan was 80%.
  • The fructan and resistant starch ratio was 15%.
  • Legume cotyledon fiber vitamin and potato fiber proportions were 5%.

In this study, the focus of our attention is whether Ancientino can improve experimental colitis and whether it can be used as a supplementary alternative to patients with ulcerative colitis. According to the cooperation agreement, the analysis data of the Ancientino belongs to the partner, and I'm really sorry we may not be provided you.

  1. Relevant Literature concerning fibers in Nutrition of humans including clinical trials have not been cited. It is quite common, that oral administration of fiber has beneficial effects on UC, therefore that should be discussed properly.

Response: Thank you for your attention. I apologize for not including more specific references to the literature on dietary fiber in my previous manuscript. Thank you for bringing this to my attention, and I hope this additional information is helpful. (see lines 480-515).

Due to its distinct function, dietary fiber is referred to as the "seventh nutrient." Dietary fiber cannot be digested or absorbed in the body [5]. Still, intestinal flora can use it as a fermentation substrate to produce beneficial sub-stances such as short-chain fatty acids (SCFAs) [6]. These substances impact on intestinal health by modifying the gut environment and permeability. Preclinical animal models have widely demonstrated the positive results of fiber supplements. Fermented barley and soybean were utilized as dietary fiber substitutes in enteritis mice. The study demonstrated that the barley-soybean combination elevated tight junction protein levels, reduced FITC-dextran permeability, and maintained the intestinal barrier's integrity [7]. Another study discovered that the mucosal pathogen C. rodentium induced fatal coli-tis in fiber-free mice, while fiber consumption improved colonic inflammation and weight loss in fiber-rich animals [8]. In the DSS-induced colitis-associated colon carcinogenesis (CAC) model, Nishiguchi et al. found that a high-fructose diet exacerbated chronic colitis. However, they also found that a fiber-rich diet protected mice from chronic colitis tumorigenesis [9]. In clinical trials, dietary fiber supplementation has shown promising results in patients with UC. A pilot research looked at how TNF-α, IL-6, and IL-8 serum levels in UC patients with remission were affected by germinated barley foodstuff (GBF). Patients in the treatment group got normal medication therapy and 30 g of GBF daily via oral administration during the two-month study. Throughout the research, the above inflammatory factors levels fell in the GBF group relative to controls [10]. Another crossover trial involving 17 UC patients who were in remission or mild found similar results. They randomly allocated patients to one of two diets: a low-fat, high-fiber diet or a regular American diet. The results showed a low-fat, high-fiber diet suppressed inflammation response [11]. Therefore, increasing dietary fiber intake is necessary to help maintain intestinal health. Furthermore, dietary fiber has been found to provide other health advantages, such as lowering the risk of chronic illnesses such as diabetes, heart disease and some cancer [12].

Overall, the literature supports the potential benefits of dietary fiber in human nutrition and health. While specific recommendations for fiber intake may vary de-pending on individual needs and health goals. Adults are typically advised to ingest at least 20-35 grams of fiber per day from a range of sources [1,2], such as grains, fruits and vegetables. However, because the fiber content of fresh fruits and vegetables is insufficient to meet this assumption, Ancientino might be utilized as a supplemental source of dietary fibers in the human diet. In this study, Mice receiving Ancientino (1200 mg/kg/day) is equivalent to ~10 g of dietary fiber daily in humans (calculation based on the ratio of mouse to human surface area). Ancientino has up to 10 g of dietary fiber in each packet. Therefore, two packets per day can adequately compensate for a shortage of dietary fiber.
